# Proposal of a Cascade Photonic Crystal XOR Logic Gate for Optical Integrated Circuits with Investigation of Fabrication Error and Optical Power Changes

**Ahmad Mohebzadeh-Bahabady [1] and Saeed Olyaee [2],\***

[1] Faculty of Electrical Engineering, Shahid Rajaee Teacher Training University, Tehran 16788-15811, Iran; a.moheb@sru.ac.ir
[2] Nano-Photonics and Optoelectronics Research Laboratory (NORLab), Shahid Rajaee Teacher Training University, Tehran 16788-15811, Iran
\* Correspondence: s_olyaee@sru.ac.ir; Tel.: +98-21-2297-0030

**Abstract:** A compact and simple structure is designed to create an all-optical XOR logic gate using a two-dimensional, photonic crystal lattice. The structure was implemented using three waveguides connected by two nano-resonators. The plane wave expansion method was used to obtain the photonic band gap and the finite-difference time-domain method was used to investigate the behavior of the electromagnetic field in the photonic crystal structure. Examining the high contrast ratio and high-speed cascade, all-optical XOR on a chip, the effects of fabrication error and the changes in the input optical power showed that the structure could be used in optical integrated circuits. The contrast ratio and data transfer rate of the cascade XOR logic gate were respectively obtained as 44.29 dB and 1.5 Tb/s. In addition, the designed structure had very small dimensions at 158.65 $\mu m^2$ and required very low power to operate, which made it suitable for low-power circuits. This structure could also be used as a NOT logic gate. Therefore, an XNOR logic gate can be designed using XOR and NOT logic gates.

**Keywords:** photonic crystal; all-optical logic gate; contrast ratio; data transfer rate; cascade

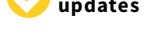

## 1. Introduction

Logic gates are devices that perform logical operations on one or more inputs and eventually generate one or more outputs. Logic gates are used to build integrated circuits. All-optical logic gates have attracted the attention of researchers due to their application in all-optical processing, the realization of all-optical computers, and the development of ultra-fast communication networks. In recent years, with the introduction of photonic crystals as suitable substrates for the design of optical devices [1–4], designing all-optical logic gates with ultra-small sizes, low power consumption, and very high switching speed for use in optical integrated circuits has become possible [5–9].

Ghadrdan et al. provided a nonlinear, effect-based structure for AND and XOR logic gates [10]. The size and delay time were 168 $\mu m^2$ and 1.34 ps, respectively. The contrast ratios for AND and XOR logic gates were 12.78 and 5.66 dB, respectively. In 2013, another logic gate was designed by using the interference effect [11]. No nonlinear materials were used in the structure, and the size and contrast ratio were equal to 620 $\mu m^2$ and 21 dB, respectively. Using a nonlinear effect on photonic crystal nano-resonators, Mohebbi et al. designed two structures for NOT, XOR, XNOR, and NAND logic gates [12]. The contrast ratios for NOT and XOR logic gates were 20.7 and 25 dB. The minimum optical power required for the input signal was 277 mW/$\mu m^2$. In 2016, OR, XOR, NOT, and AND logic gates were designed using a ring resonator [13]. The proposed structures for NOT and XOR logic gates operated at three wavelengths of 1512, 1527, and 1544 nm, and the contrast ratio at best was 23.04 dB. A structure was introduced for various logic gates including AND,

NOT, and XOR, which worked based on the interference effect [14]. The size and contrast ratio of this structure were 155 μm$^2$ and 15.18 dB, respectively. Olyaee et al. presented all-optical NOT and XOR logic gates based on the interference effect with high contrast ratios and low footprints. The dimensions of the proposed structure, contrast ratio, and response time were reported as 85 μm$^2$, 43.38 dB, and 0.317 ps, respectively [15]. Comparing the structures mentioned above, it can be concluded that not using non-linear materials greatly reduces the optical power required by the structure and increases the data transfer speed. In addition, using nano-resonators instead of nano-ring resonators makes the structure smaller and easier to implement.

In this paper, a structure is designed to be used as an all-optical XOR logic gate, and the performance of the XOR logic gate in optical integrated circuits is investigated. The design of the structure was very simple and consisted of three waveguides connected by a T-shaped connector. The logic gate was designed based on the interference effect. In addition to the structure design, the parameters that measure the performance of the designed logic gate in optical integrated circuits were investigated. Moreover, the manners in which two logic gates were placed consecutively (in cascade form) in a logic circuit, the effect of fabrication error, and the changes in input optical power were investigated. The results showed that the designed logic gate could be effectively used in low-power optical integrated circuits.

The rest of the paper is organized as follows: In the second part, important parameters for evaluating the performance of logic gates are presented. In the third and fourth sections, the designed structure and its operation are respectively described. The efficiency of the logic gates in the optical integrated circuits is investigated in Section 5. Finally, the conclusion is drawn in Section 6.

## 2. Important Parameters in Evaluating the Efficiency of All-Optical Logic Gates

The efficiency of optical logic gates is evaluated and compared using several parameters [16]. One of the important parameters is contrast ratio. This parameter indicates the difference between logic levels "1" and "0" and can be defined as:

$$CR = 10 \log \left( \frac{P_{\text{on}}}{P_{\text{off}}} \right) \tag{1}$$

where $P_{\text{on}}$ is the minimum optical power for logic "1", and $P_{\text{off}}$ is the maximum optical power for logic "0".

Another parameter is the response time of the logic gate, which is a factor in the operating speed of that gate. With a shorter response time, faster data transmission is possible. By monitoring the output of the logic gate and comparing the time of the two input and output signals, the response time can be obtained [17]:

$$t_d = \frac{t_r + t_f}{2} \tag{2}$$

where $t_r$ and $t_f$ are the rise time and the fall time, respectively. In all-optical logic gates, it can be said that the rise time will always be longer than the fall time [18]. Considering the propagation delay time ($t_p$), the response time can be given by:

$$t_R = t_p + t_d \tag{3}$$

Therefore, the arrival time of the output power from the beginning of the input signal to the structure until the output power reaches 90% of the steady-state power is calculated as the response time of the logic gate [19,20].

The footprint and the amount of optical power or output intensity [21] are also important factors in evaluating the performance of logic gates. Because the purpose of designing logic gates is to achieve all-optical integrated circuits, designing a logic gate with a smaller size is more suitable for integrated circuits. In addition, in all-optical logic

gates, if the output intensity is less than a suitable level, the next logic gate will lose its efficiency. Therefore, the output intensity in a logic state must be approximately within the same range of the input optical power; otherwise, a detection error occurs.

## 3. The General Structure of the Logic Gate's Parameters

To design this logic gate, a photonic crystal structure with a square lattice was considered. The lattice constant, radius, and height of the dielectric rods were 519 nm, 100 nm, and 220 nm, respectively. The photonic crystal structure used consisted of an array of 14 × 20 dielectric rods made of silicon. According to the photonic crystal structure and the contrast ratio, the footprint of the XOR logic gate was 66.53 μm². According to the band structure, which was obtained by the plane wave expansion (PWE) method, the band gap of the structure was at wavelengths from 1210 nm to 1782 nm in transverse magnetic (TM) mode. To analyze the behavior of the electromagnetic field in the structure, the finite difference time-domain (FDTD) method was used. Although the structure analysis required three-dimensional analysis, it needed a long calculation time and a high amount of memory. If the height of the dielectric rods was large enough, two-dimensional analysis could perfectly cover the behavior of the structure [22]. The spatial separations $\Delta x$ and $\Delta y$ needed to be small enough to ensure that the calculations were correct:

$$\Delta x = \Delta y \leq \frac{a}{20} \tag{4}$$

In this paper, simulations with a mesh size of $\Delta x = \Delta y = a/32$ were selected. The following condition needed to be met to obtain a stable simulation:

$$\Delta t \leq \frac{1}{c} \frac{1}{\sqrt{\frac{1}{(\Delta x)^2} + \frac{1}{(\Delta y)^2}}} \tag{5}$$

where $\Delta t$ represented the time step for the simulation, which was selected as $\Delta t = \Delta x/2c$. To obtain steady-state results, five iterations for the FDTD method were used.

The structure designed to be realized for the XOR logic gate is shown in Figure 1. The structure consisted of three waveguides and two nano-resonators, located at the T-shaped junction. The size of nano-resonators was selected as 30 nm. At this size, the best optical transmission from the input waveguides to the output occurred, and the output power in the equivalent state had the maximum value. I1 and I2 were for receiving the input signal, and O1 was for receiving the output signal. If this structure was used as a NOT logic gate, one of the input ports was assigned to the control signal, which always entered the optical power equivalent to one into the structure.

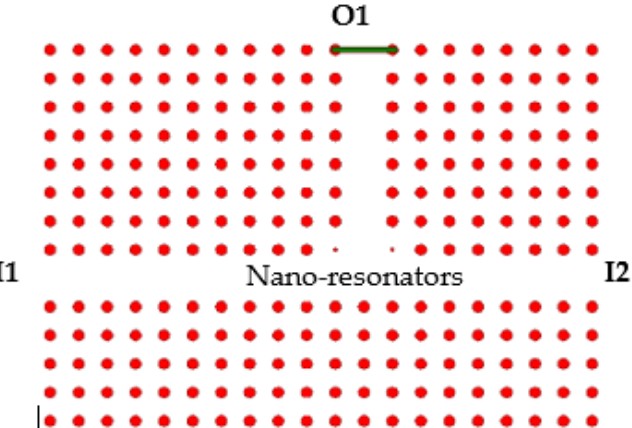

**Figure 1.** Structure designed to realize an all-optical XOR logic gate based on a photonic crystal with a square lattice.

## 4. Assessing the Performance of the Logic Gate

The performance of the all-optical XOR logic gate was assessed in three cases. In the first case, only the optical source placed in port I1 was active. Due to the lack of signal at the opposite port, the light entering the structure reached the output waveguide by transmitting it from the nano-resonator, and the output was at logic level "1". The optical power output, in this case, was equal to 0.744 $P_{in}$. In the second case, only the optical source placed in port I2 was active. In this case, the power reached the output by passing through the nano-resonator, and 74.0% of the input power reached the output port. The third case was related to the activation of both optical sources. When the input signals reached the two nano-resonators, they interfered destructively, and only a very small amount of input power was detected at the output port. The magnetic field patterns in these three cases are shown in Figure 2.

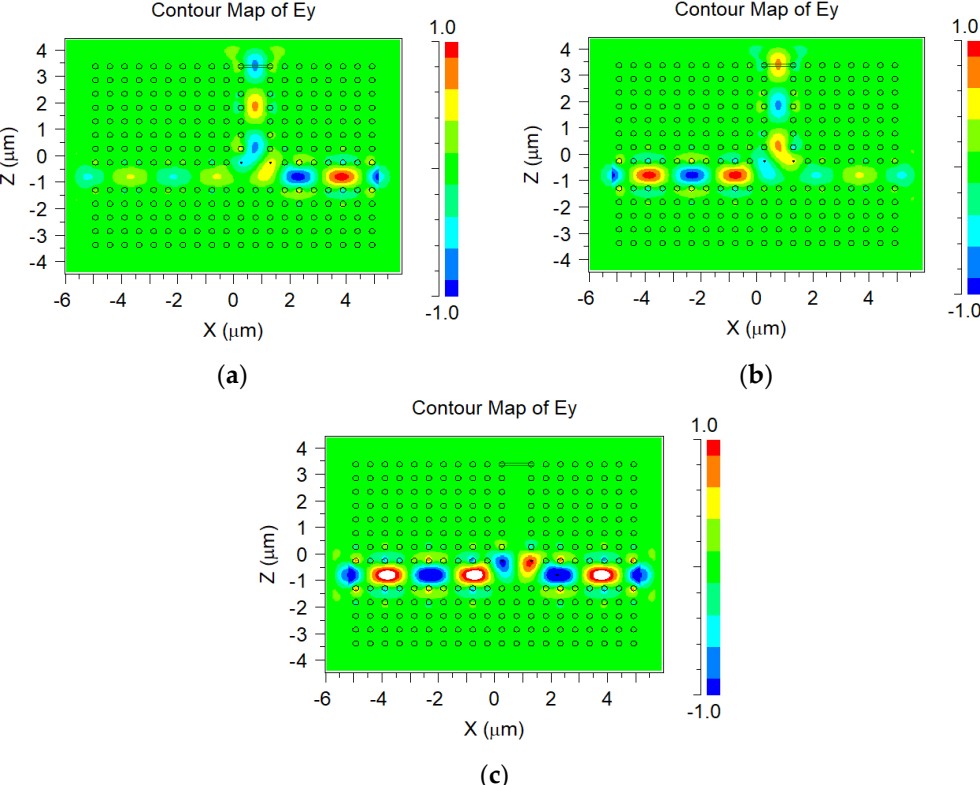

**Figure 2.** Magnetic field patterns of all-optical XOR logic gates in input logic mode: (**a**) 01, (**b**) 10, and (**c**) 11.

These three cases perfectly displayed the performance of the designed logic gate for the XOR logic gate. To use this structure for the NOT logic gate, the third case could be used with either the first or second cases. The delay time and data transmission rate were equal to 0.317 ps and 3.15 Tb/s, respectively. The truth table and the power output levels of the all-optical XOR logic gate are given in Table 1. According to the amount of output optical power in the logic states "1" and "0", the contrast ratio was equal to 43.38 dB, which was a high value.

**Table 1.** Truth table and power output levels of all-optical XOR logic gate.

| Input Power A | Input Power B | Output Power $\times\ 10^{-3}$ |
|---|---|---|
| 0 | 0 | 0 |
| 0 | $P_{in}$ | $740P_{in}$ |
| $P_{in}$ | 0 | $744P_{in}$ |
| $P_{in}$ | $P_{in}$ | $0.034P_{in}$ |

## 5. Investigating the Efficiency of XOR Logic Gates in Optical Integrated Circuits

In order to use all-optical logic gates in optical integrated circuits, in addition to the contrast ratio and delay time, other parameters must be considered, and the performance of the structure must be examined accordingly. These parameters include how the logic gates are placed together, the fabrication error, the amount of power consumed, and the changes in the optical power input. In this section, these parameters are reviewed.

### 5.1. Cascade All-Optical XOR Logic Gate

To evaluate the performance of the XOR logic gate in optical integrated circuits, the cascade XOR logic gate was investigated. To assess this issue, a structure consisting of two consecutive logic gates was placed according to Figure 3, and by applying a signal to the input ports, the desired output was received. The structure consisted of three input ports and one output port. Inputs I1 and I2 were for the first logic gate. The output of the first logic gate and input I3 were also considered the inputs of the second logic gate. According to Figure 3 and considering a 519 nm lattice constant, the footprint of cascade XOR logic gate was 158.65 µm².

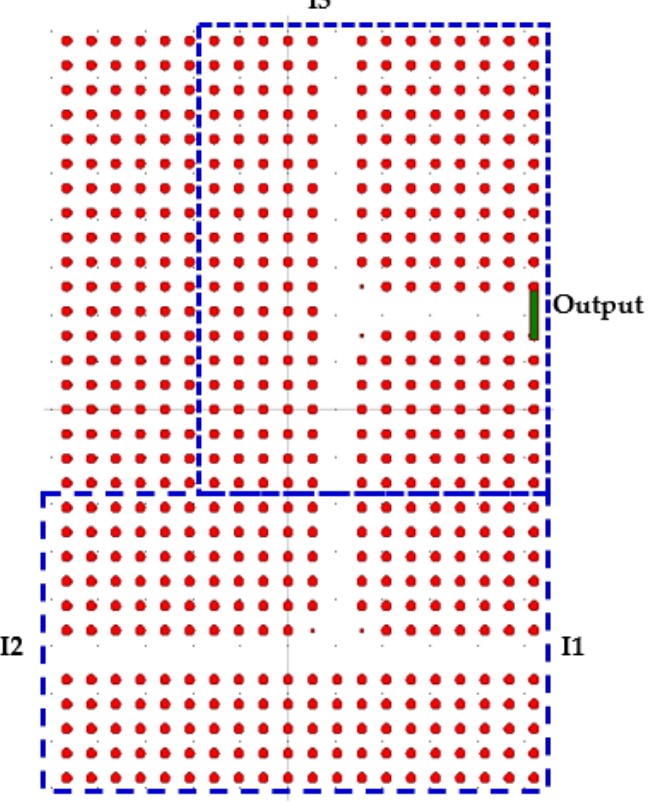

**Figure 3.** Structure designed to investigate the effects of placing two all-optical XOR logic gates consecutively. This structure could be used for all-optical XNOR logic gates.

The study was performed in two general cases. The behavior of the electromagnetic fields is shown in Figures 4 and 5. These two cases were related to the activation or deactivation of the optical source located at input I3. By activating or deactivating this optical source, four cases related to the first XOR logic gate were investigated, and the output in the structure was obtained.

According to the pattern of the electromagnetic field, it could be concluded that the cascade XOR logic gate worked well. As a result, the XOR logic gates could be used in optical integrated circuits. The truth table and power levels are given in Table 2. The value of the output power in a state equal to logic "1" indicated that in this gate, after two uses, the output power was within acceptable limits and could easily be used as the input of the logic gate of the next level.

The structure used to examine the sequential placement of two logic gates could be considered in another way. The first logic gate needed to be considered the XOR logic gate, and the second logic gate needed to be considered the NOT logic gate. In this case, ports I1 and I2 applied input signals to the XOR logic gate, and port I3 applied the control signal to the NOT logic gate.

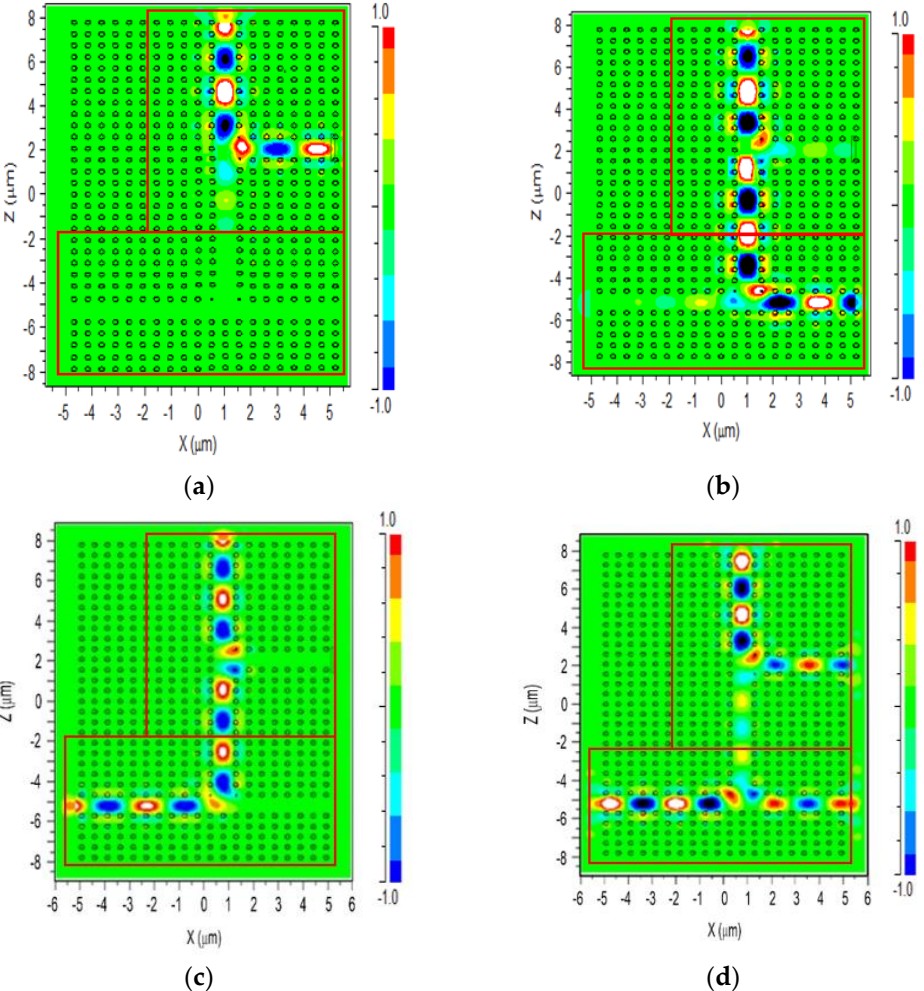

**Figure 4.** Behavior of the electromagnetic field for different input states in the all-optical XNOR logic gate in the active state of the control signal. (**a**) Both input signals were inactive, (**b**) the input signal of port 1 was active and in port 2, inactive, (**c**) the input signal of port 1 was inactive and in port 2, active, and (**d**) both input signals were active.

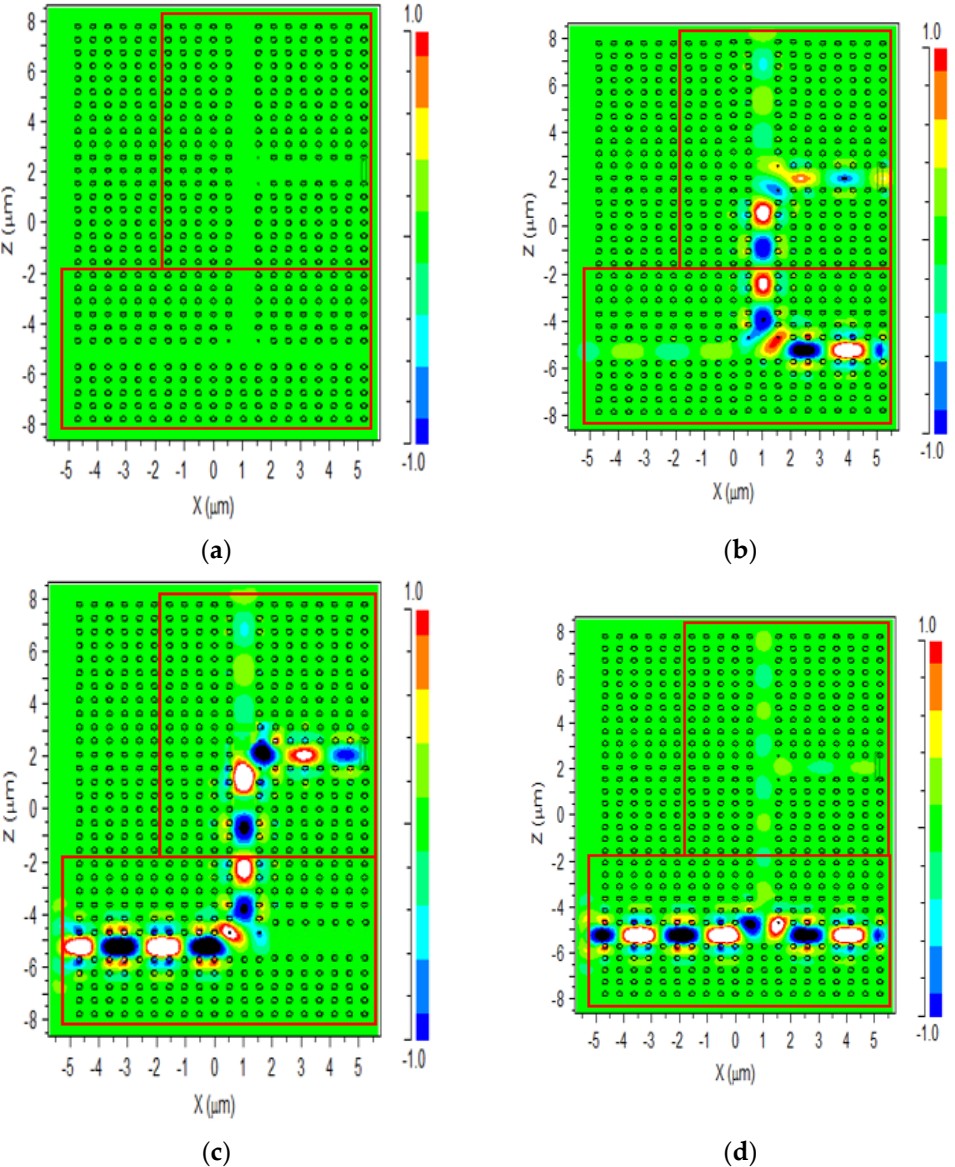

**Figure 5.** Behavior of the electromagnetic field for different input states in the all-optical XNOR logic gate in the inactive state of the control signal. (**a**) Both input signals were inactive, (**b**) the input signal of port 1 was active and in port 2, inactive, (**c**) the input signal of port 1 was inactive and in port 2, active, and (**d**) both input signals were active.

As could be seen from the behavior of the electromagnetic field, if the control signal was active, the optical power received at the output port was the inverse of the optical power of the XOR logic gate. Therefore, the structure proposed in this section had the function of the all-optical XNOR logic gate. If the control signal was off, the optical power received at the output port was the same as the optical power of the lower logic gate. Therefore, the proposed structure displayed the all-optical XOR logic gate. As a result, this structure could be considered a structure that had two operators, XNOR and XOR, with the control signals ON and OFF. The truth table of these two logic gates is given in Table 2. When input I3 was inactive, the XOR logic gate was formed and when I3 was enabled, the XNOR logic gate was formed.

This study investigated the use of the structure designed as the XOR logic gate. For the cascade XOR logic gate, the contrast ratio, delay time, and data transfer rate were respectively obtained as 44.29 dB, 0.567 ps, and 1.5 Tb/s. Moreover, the results were valid

for the NOT logical gate. To investigate the sequential placement of two NOT logic gates, only modes B and D of Figure 4 should be considered.

**Table 2.** Truth table and power levels of the all-optical XNOR logic gate in two modes in the inactive/active states of the control signal.

| Input Power I3 | Input Power I2 | Input Power I1 | Output Power $\times 10^{-3}$ |
|:---:|:---:|:---:|:---:|
| 0 | 0 | 0 | 0 |
| 0 | 0 | $P_{in}$ | $618P_{in}$ |
| 0 | $P_{in}$ | 0 | $621P_{in}$ |
| 0 | $P_{in}$ | $P_{in}$ | $0.023P_{in}$ |
| $P_{in}$ | 0 | 0 | $728P_{in}$ |
| $P_{in}$ | 0 | $P_{in}$ | $0.023P_{in}$ |
| $P_{in}$ | $P_{in}$ | 0 | $0.023P_{in}$ |
| $P_{in}$ | $P_{in}$ | $P_{in}$ | $709P_{in}$ |

### 5.2. Investigating the Effects of Fabrication Error

In general, two types of photonic crystals can be designed for devices based on these components. The first type is based on air holes in a dielectric bed, and the second type is based on dielectric rods located in the air [23–27]. While it is argued that photonic crystals from air holes are relatively simple to make using lithography and etching methods, recently, designing photonic crystal devices based on dielectric rods has become more attractive using bottom-up fabrication techniques, which have the advantage of being able to place the desired features in exact locations [22]. To investigate the fabrication error of the logic gate, several zones in the structure that had a greater impact on the performance of the logic gate were considered. As shown in Figure 6, three zones were considered for this study. The first zone (Zone 1) was the area around the nano-resonators. The second zone (Zone 2) was related to the dielectric rods around the waveguides. Finally, the third zone (Zone 3) was an area away from the waveguides.

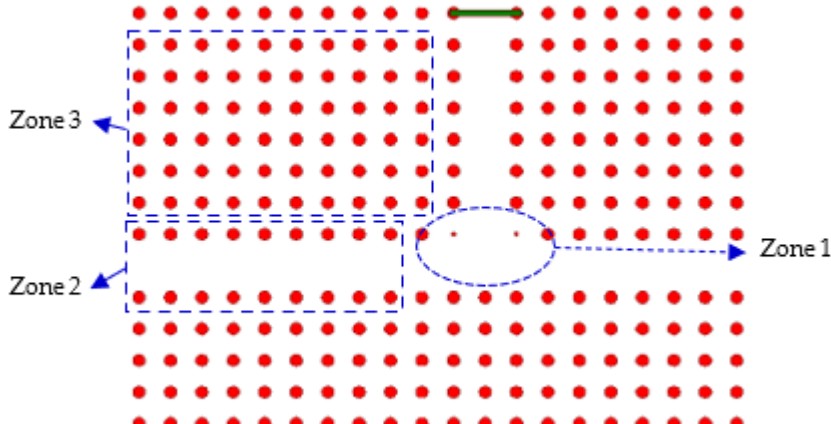

**Figure 6.** Showing important zones to investigate the effects of fabrication errors on the all-optical XOR logic gate.

The first state was when the rods for the nano-resonator did not form the desired size during fabrication. The output power in the two possible modes for the output (logic levels "1" and "0"), based on changing the radius of the nano-resonator rods from 10 to 50 nm, is shown in Figure 7. In these figures, the horizontal axis is a different size for the nano-resonator radius, in nanometers, and the vertical axis is the output optical power. If the changes in the size of the dielectric rod were greater than 40 nm or less than 20 nm, the intensity of the output power declined and consequently, the contrast ratio value

decreased. According to Figure 7a, in the case where the nano-resonator size was 30 nm, most of the electromagnetic field was transmitted from the input port to the output port. For other sizes, the amount of light transmission to the output port was less. This decrease was due to the fact that the light did not couple well between the two waveguides. In Figure 7b, the light entering from the two input ports has destructive interference with the nano-resonator and receives the least amount in the output port. As shown in Figure 7b, for the best case, the radius of nano-resonator was equal to 30 nm.

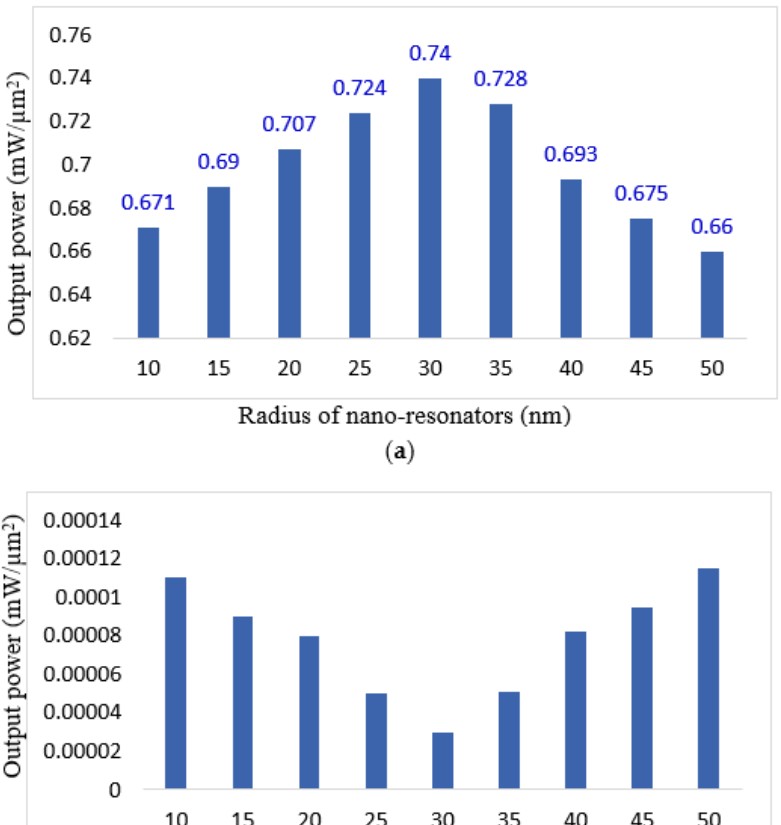

**Figure 7.** Output power of the all-optical XOR logic gate, proportional to the changes in the radii of nano-resonators from 10 to 50 nm in different input states for (**a**) 01 and (**b**) 11.

The second state was changing one or more rods around the waveguide. The simulations performed in this case led to the conclusion that if these changes were not large (up to 12%), they would not affect the correct operation of the logic gate. However, if this error was closer to the location of the nano-resonator rods, the fabrication error was greater. If this error was exactly around the nano-resonator, due to the fact that the nano-resonator itself was formed by reducing the size of the dielectric rod, any decrease or increase in the size of the side dielectric rod or any deformation resulting from its fabrication error was known as a kind of resonator and disrupted the function of the structure. The simulation in this particular state showed that with only a few percentage changes (up to 5%) in the size of the dielectric rod, the designed logic gate performed its function correctly. If the amount of changes was more than this value, the intensity of the output power was reduced and as a result, the logic gate lost its efficiency.

The third state was to investigate the effects of fabrication error on locations away from waveguides. Changes at these points had almost no effect on the performance of the logic gate structure. If the location of the error was far enough from the waveguides and nano-resonators (three rows away from the nano-resonators and the output and input ports

was enough distance), even not creating a dielectric rod did not affect the performance of the logic gate.

### 5.3. Investigating the Effects of Changing the Input Optical Power

In this study, only two different modes of XOR logic gate performance were investigated. In other words, this study was conducted for evaluating the performance of the structure for the NOT logic gate, which could be generalized to the XOR logic gate. The optical power used for the input signal and the control signal sources in the initial design was $1 \, \text{mW}/\mu\text{m}^2$. To check the efficiency of the logic gate at the following powers, the amount of optical power related to these two sources was changed in the range of 0.2 to $1 \, \text{mW}/\mu\text{m}^2$, and the amount of output optical power was calculated. This study was performed with two cases. The first case involved changing both sources used in the structure. Changing the optical power of the input signal source by keeping the optical power constant in the control signal occurred in the second case.

The results of the simultaneous change of power in two sources (the first case of the study) showed that when decreasing the value of the input power, the output power also decreased. Because nonlinear materials were not used, and there was no need for starting power in the structure, with changes in input power, the output power changed correspondingly. Therefore, the ratio of output power to input power in different simulation modes was almost constant. The same power ratio for optical power changes kept the contrast ratio constant. Therefore, the designed logic gate with different optical powers had its own efficiency and could be a good option for use in low-power optical integrated circuits.

Due to the fact that the control signal was connected to a separate optical source, it was unlikely to change its power value unless the entire logic gate was used in low-power circuits. In contrast, using this logic gate in integrated optical circuits might reduce the optical power of the input signal because the optical power might be provided by another level. To investigate this condition (the second case of the study), the optical power of the control signal was kept constant and equal to $1 \, \text{mW}/\mu\text{m}^2$, and the optical power of the input signal was changed from 0.2 to $1 \, \text{mW}/\mu\text{m}^2$. The values for the output optical power in both logic states "1" and "0" are given in Figure 8.

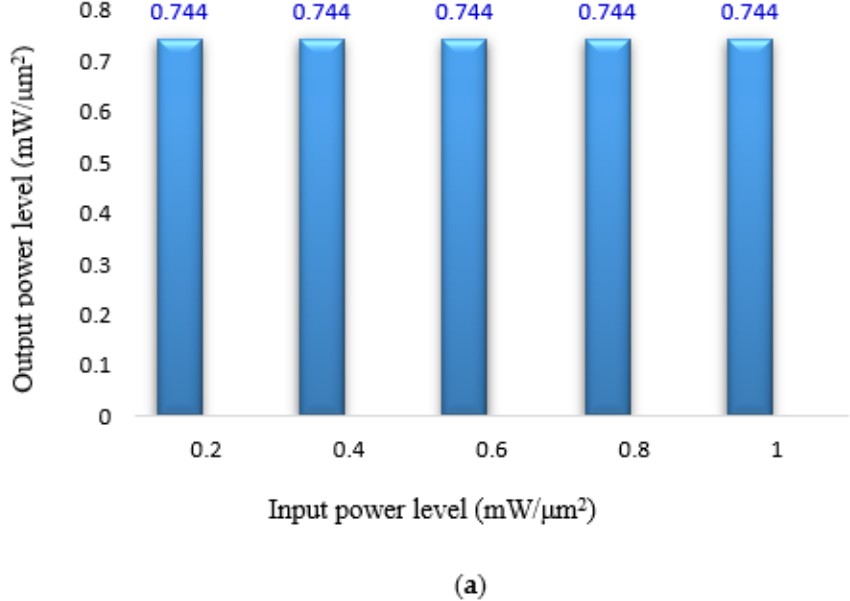

**(a)**

**Figure 8.** *Cont.*

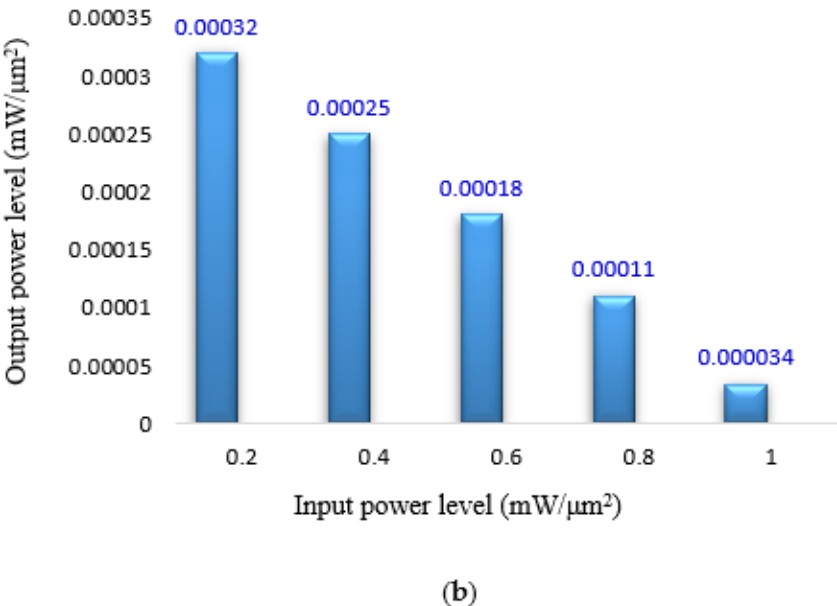

**(b)**

**Figure 8.** Output optical power for different values of input optical power for designed XOR logic gate; (**a**) output power in one logic mode and (**b**) output power in zero logic mode. In this study, the optical power of the control signal was constant, and only the power of the input signal changed.

When the input signal was off, due to the constant value of the control signal power, the output power was the same and equal to 3 (Figure 8a). When the input signal was on, due to the decrease in the amount of optical power input compared with the control signal in different modes of investigation, the amount of power reaching the output port was greater than the equal state of the input and control signals (Figure 8b). This increase in the output power of logic "0" reduced the amount of contrast ratio, which affected the efficiency of the logic gate to some extent.

Table 3 compares the designed single-XOR structure with previous structures. The designed structure was smaller than previous works and offered a higher contrast ratio. In addition, the output power level in logic level "1" was large enough that it could easily activate the logic gates of the next level. These advantages as well as the lack of high optical power made the all-optical XOR logic gate suitable for use in optical integrated circuits.

**Table 3.** Comparison of the results of the designed all-optical XOR logic gate with the previous designs.

| | Crystal Lattice Type | Contrast Ratio (dB) | Power at Zero Logic Value (mW/μm²) | Power at One Logic Value (mW/μm²) | Delay Time (ps) | Size (μm²) |
|---|---|---|---|---|---|---|
| This paper | Square | 43.38 | $3.4 \times 10^{-5}$ | 0.740 | 0.317 | 66.53 |
| [10] | Square | 5.67 | 0.22 | 0.81 | 0.85 | 168 |
| [11] | Triangular | 21.44 | 0.0061 | 0.85 | – | 620 |
| [14] | Square | 12.155 | 0.0182 | 0.535 | – | 155 |
| [28] | Triangular | 19.28 | 0.0078 | 0.776 | 0.466 | 136 |
| [29] | Triangular | 6.50 | 0.133 | 0.574 | | – |
| [30] | Square | 55.23 | $1.2 \times 10^{-6}$ | 0.4 | 0.136 | 106 |
| [31] | Triangular | 8.95 | 0.07 | 0.55 | – | 729 |
| [32] | Triangular | 30 | – | – | 0.10 | 105 |

### 6. Conclusions

In this paper, photonic crystal structures were proposed to be used as all-optical XOR and NOT logic gates. The logic gate mechanism was based on the interference effect and was designed using a photonic crystal nano-resonator. Contrast ratio, delay time, and data transfer rate for this structure were 43.38 dB, 0.317 ps, and 3.15 Tb/s, respectively. In this paper, in addition to designing the structure, the efficiency of the logic gate in optical integrated circuits was investigated. For the cascade XOR logic gate, the fabrication error and the effects of optical power change at the input were examined. Considering the cascade form, the contrast ratio, delay time, and data transfer rate were 44.29 dB, 0.567 ps, and 1.5 Tb/s, respectively. According to the results, due to the simplicity of the structure and its low power consumption, this logic gate is suitable for use in low-power optical integrated circuits.

**Author Contributions:** A.M.-B. designed and performed simulations and analyzed data and S.O. supervised, edited, and prepared the final draft of the manuscript. All authors have read and agreed to the published version of the manuscript.

**Funding:** This research received no external funding.

**Institutional Review Board Statement:** Not applicable.

**Informed Consent Statement:** Not applicable.

**Data Availability Statement:** Data sharing is not applicable to this article.

**Acknowledgments:** This work was done in the Nano-photonics and Optoelectronics Research Laboratory (NORLab), Shahid Rajaee University.

**Conflicts of Interest:** The authors declare that they have no known competing financial interest or personal relationships that appear to influence the work reported in this paper.

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
