# Peer review of "Proposal of a Cascade Photonic Crystal XOR Logic Gate for Optical Integrated Circuits with Investigation of Fabrication Error and Optical Power Changes"

_photonics, doi:10.3390/photonics8090392_

Round 1
Reviewer 1 Report
In this paper, the authors designed a simple structure for all-optical XOR logic gates with a two-dimensional photonic crystal lattice. The effect of fabrication error and changes in the input optical power also studied. This article is clear, concise, and suitable for the scope of the journal. Several suggestions are supplied:
- Suggest improving the resolution of Fig.4 (a d) and Fig.7.
- Suggest give more detail about the decrease and increase slope performance in Fig.7.
- Please make references uniform, please check.
Reviewer 2 Report
The paper is well written and the results are clearly presented.
Reviewer 3 Report
The manuscript “Investigation of fabrication error and optical power changes for cascade all-optical XOR logic gate in optical integrated circuits” by Ahmad Mohebzadeh-Bahabady and Saeed Olyaee deals with an all-optical XOR logic gate and the influence of fabrication error and optical power changes.
This topic is interesting for publication in Photonics.
However, several aspects of the manuscript need improvements before recommending it for publication.
The language of the manuscript needs major improvements.
1. Abstract: Skip “In this paper,…” as starting phrase. Middle: “that the structure can to be used in optical integrated” End: “and an XNOR logic gate”
2. Introduction: Second paragraph describes what was done in References 10 to 15 and all the sentences have the same structure. This paragraph needs rewriting. It should include the main advances in technology that made the realization of several structures possible. Advantages and disadvantages should be briefly discussed. Otherwise it looks like an arbitrary list of results.
3. Section 4 /Page 4: 0.740% should be 74.0%
4. Table 1: It should be P_in.
5. Section 5.2: It is the first section dealing with fabrication error. This is rather late since it is the first words of the title of the manuscript. So the title is rather misleading. It should be changed to represent the major aspects of the manuscript.
6. Figure 6: The areas should be labeled also with 1-3, according to the discussed cases.
7. Figure 8: The results of Figure 8 are rather obvious. Figure 8 is not necessary. “Since nonlinear materials are not used and there is no need for starting power in the structure, with changes in input power, the output power changes correspondingly. Therefore, the ratio of output power to input power in different simulation modes is almost constant. The same power ratio for optical power changes keeps the contrast ratio constant. Therefore, the designed logic gate with different optical powers has its own efficiency and can be a good option for use in low-power optical integrated circuits”
8. Conclusion: “For cascade XOR logic gate, the fabrication error and the effects of optical power changes at the input have been examined.” Only one sentence does not match the role in the title. The title should be changed.
Round 2
Reviewer 1 Report
The authors replied to my comments, so now I can recommend it publish on Photonics.
Reviewer 3 Report
The authors have improved the manuscript according to the suggestions by the reviewers.